
# Noise in raw data of magnetic observatories

Sergey Y. Khomutov[1], Oksana V. Mandrikova[1], Ekaterina A. Budilova[1,2], Kusumita Arora[3], and Lingala Manjula[3]

[1]Institute of Cosmophysical Research and Radio Wave Propagation FEB RAS, Mirnaya str, 7, Paratunka 684034, Kamchatka, Russia

[2]Kamchatka State Technical University, Klyuchevskaya str, 35, Petropavlovsk-Kamchatsky 683003, Russia

[3]CSIR- National Geophysical Research Institute, Uppal Road, Hyderabad-500007, Telangana, India

*Correspondence to*: Sergey Y. Khomutov (khomutov@ikir.ru)

**Abstract.** In spite of significant progress in the development of new devices for magnetic measurements, mathematical and computational technologies for data processing and means of communication, the quality of magnetic data accessible through the world centers still largely depends on the actual conditions in which observation of the Earth's magnetic field is performed at observatories. Processing of primary results of magnetic measurements by observatory staff plays an important role. It includes effective identification of noise and elimination of its influence on final data. In this paper, on the basis of the experience gained during long-term magnetic monitoring carried out at the observatories of IKIR FEB RAS (Russia) and CSIR-NGRI (India), we present a review of methods commonly encountered in actual practice for noise identification and the possibilities to reduce noise influence.

## 1 Introduction

Magnetic measurements at observatories are an important source of information to study the processes in the Earth's interior and in the upper shells that substantially supplements the data obtained from satellites, during magnetic surveys or from temporary stations. Currently, about 130 magnetic observatories are integrated into the global observation network INTERMAGNET (www.intermagnet.org), which established the standards for measurements, processing and transmission of the data. INTERMAGNET standards define the requirements for technical parameters of variational and absolute observations and set requirements for the accuracy of final data, but do not specify characteristics of raw data, for example, the presence of noise which determines data quality. It is assumed that these issues are addressed at the level of individual observatories (INTERMAGNET Tech. Ref. Manual, 2012). Final data of INTERMAGNET observatories (quasi-definitive and definitive) undergo multistage control, but their validity and reliability greatly depend on the quality of the results of primary magnetic measurements.

There are a lot of reasons why the data quality decreases, for example, methodological and hardware (technical) problems, noise caused by environment, organizational difficulties, etc. Methodological issues are solved in large part by standard



requirements according to which measurements are carried out (for example, INTERMAGNET standards in INTERMAGNET Tech. Ref. Manual (2012)) or in guides for organization of magnetic measurements (Jankowski and Sucksdorff, 1996; Nechaev, 2006). Hardware problems are mainly solved by the developers at the level of production (development and production of magnetometers) and partly by a user at an observatory (calibration, comparisons, etc.). In

the result of influence of many external sources, noises manifest as signals in the magnetic field which are registered by magnetometers.  A considerable part of papers about noise in magnetic measurements is devoted to hardware noise, or noise from uncontrolled sources. The first type is oriented to the developers and in many cases developers are authors and co-authors (see, for example, Denisov et al. (2006), Hegymegi et al. (2016) and Khomutov et al. (2016). The second type, in its turn, represents almost complete scientific surveys, with the results of investigation of noise properties, physics of their

sources, etc. (see, Maule at al. (2009), Neska et al. (2013) and Santarelli et al. (2014)). At the same time, both the first, and the second types of papers are not often oriented to give practical recommendations to magnetologists carrying out primary data processing.

Therefore, it seems necessary to make a review of man-made disturbances, that are most frequently encountered in the raw

magnetic data and to show examples of possible methodological and software techniques that allow us to eliminate the noise with varying efficiency. As the initial data for the analysis, we considered the results of magnetic measurements, which are carried out at INTERMAGNET observatories: Paratunka, Magadan, Khabarovsk (Russia) and Hyderabad (India). Besides, the data of Cape Schmidt and Choutuppal observatories were used.

The topicality of the information is confirmed by the following:

1)  magnetologist generation changes, and new generation is sure that all possible problems at the observatories can be solved by modern technologies. At the same time, the lack of experience of real work at magnetic observatories and the absence of full information about actual conditions of measurements can lead to serious negative consequences;

2) due to the specific subject (it is mainly discussed in a narrow circle of specialists, who monitor the magnetic field at the observatories), the results and the conclusions of these discussions remain outside of access through publications, and in the best case, they appear in the conference proceedings, i.e. are limited in distribution;

3) the amount of data with which scientists have to deal, has grown significantly. It is almost impossible to perform sufficiently correct estimation of the quality of these data, because we deal with the final results of measurements carried out at observatories, and there is no information indicating the nature of potential problems in this data.



The term "noise" is relative and significantly depends on the specific problems to be solved, used equipment, requirements of accuracy and time resolution, and so on. For example, temporary signals in the magnetic field caused by the sources in the ionosphere-magnetosphere are considered as noise during ground magnetic survey and interpretation of its results. Concerning the magnetic measurements at INTERMAGNET observatories, the signals, which have sources closer than a few

tens of kilometers, can be conventionallyconsidered as noise (see, for example, Santarelli et al., (2014)). Of course, there are some powerful sources of man-made noise such as DC railways, which in case of appropriate conductivity of the upper layer of the Earth's crust can produce a significant effect in the magnetic data at large distances. On the other hand, some natural phenomena can have a smaller spatial sizes than the limit value given above. For example, tectonomagnetic and seismomagnetic effects with typical distances to the suspected source of tens of kilometers.

An indication for recognition of signal as noise can be the typical duration of this signal. We can be assume that this time is generally not significant, for example, about one hour and shorter. At the same time, it is necessary to keep in mind that there are many natural signals with the considered durations. There are also many examples of random noise signals with the spectrum similar to natural signals, which can last from days up to months. However, even such random noise often

represents a mixture (superposition) of many signals which have much shorter characteristic time of existence (seconds or minutes) and clear structure. For example, railway signals in magnetic measurements have pulse or rectangular shape at short distance from railway (Neska et al., 2013). Therefore, the efficiency of time criterion for estimation of noise is low.

In this work we will consider only the noises, the man-made nature of which has already been proved or its structure allows

us to interpret them unambiguously. Magnetic signals with features of noise, but with unknown sources, will be out of scope of this work. The data used in this paper were processed by the tools of MATLAB mathematical software package (www.mathworks.com) and by application software applied in MATLAB and Octave (http://www.gnu.org/software/octave/) environments used at observatories.

## 2 Initial data, description of observatories

In this paper we apply the data collected during regular magnetic measurements at the observatories of IKIR FEB RAS (Russia) and CSIR-NGRI (India). The observatories are listed in Table 1, their location is shown in Fig. 1. Table 1 shows the name of each observatory, IAGA code, geographical coordinates, institute and status in INTERMAGNET network, and magnetometers.

All listed observatories, with the exception of HYB, are located far enough from big cities, but in the vicinity of small settlements. There are no powerful sources of potential noise, such as factories, railways and etc. nearby (up to 10-30 km). Hyderabad observatory (HYB) is located within the city and an above-ground subway line has been recently built a few





hundreds of meters from the pavilions. Russian observatories CPS, MGD, PET and KHB were built in the 60s according to the requirements for complex magnetic-ionospheric stations in the USSR, with a wide range of geophysical observations. In the result, there are other observation systems in the immediate vicinity of magnetic pavilions, which are potential sources of interference, for example, ionosondes for vertical sounding of the ionosphere. Moreover, due to the limited area, there are

5   some facilities such as garages with heavy machinery, wells, electricity power equipment etc. near the observatories which may influence the magnetic measurements.

**Table 1. List of magnetic observatories, the data of which are used in this article. The magnetometers are marked by normal font for vector devices and italic font for scalar ones. IMO is INTERMAGNET magnetic observatory.**

| Observatory | IAGA | Lat(N) | Lon(E) | Institute | Magnetometers |
|---|---|---|---|---|---|
| Cape Schmidt | CPS | 68.9 | 180.6 | IKIR | dIdD, Magdas, *POS-1* |
| Magadan | MGD | 60.1 | 150.7 | IKIR, IMO | FGE, FRG-601, Magdas, dIdD, *GSM-90, POS-1* |
| Paratunka | PET | 53.0 | 158.2 | IKIR, IMO | FGE, FRG-601, Magdas, dIdD, POS-4, *GSM-90, POS-1* |
| Khabarovsk | KHB | 47.6 | 134.7 | IKIR, IMO | dIdD, Quartz-06, *POS-1, GSM-19W* |
| Choutuppal | CPL | 17.3 | 78.9 | NGRI | FGE, GEOMAG-02M, GEOMAG-02MO, *GSM-90, GSM-19W* |
| Hyderabad | HYB | 17.4 | 78.6 | NGRI, IMO | FGE, GEOMAG-02, *GSM-90, GSM-19W* |

Extreme climate conditions at IKIR observatories are of great importance. For MGD and KHB observatories, sharply continental climate with seasonal changes of temperature from -40 to +30ºC is ordinary. CPS observatory is located in the High Arctic zone with very hard climate conditions. PET observatory is characterized by abundance of precipitation, the level of snow in winter is up to 2 m. These conditions require special and expensive means to provide the required

15   temperature conditions in magnetic pavilions. Special machinery is necessary to clear the ways to the pavilions from snowthat may also affect the quality of measurements. We should mention such features as stability of external power supply that is difficult to provide at the observatories remote from densely populated areas, as well as ground connection quality problems due to the nature of ground (at Cape Schmidt and Magadan). For Indian observatories climate conditions are also a significant problem, high heat during the year and the rainy season with high humidity.





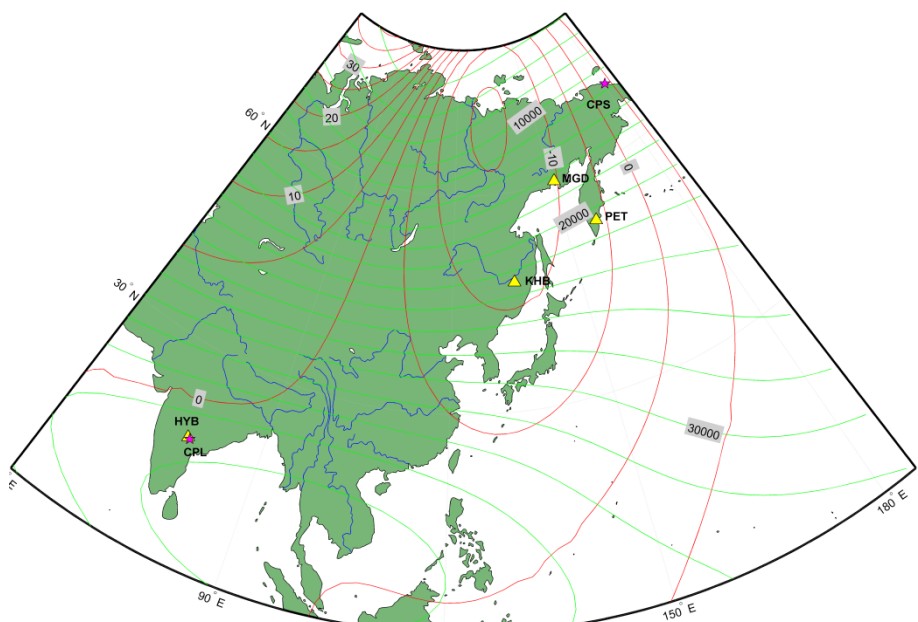

**Figure 1. Location of magnetic observatories of IKIR FEB RAS (CPS, KHB, MGD and PET) and CSIR-NGRI (HYB and CPL). Isolines show horizontal component H (green lines) and declination D (red lines) according to IGRF12 model. Yellow triangles are the INTERMAGNET observatories, red stars are non-INTERMAGNET observatories.**

## 3 Description and classification of noise in raw data

### 3.1 "Regular-random" noises

In order to develop a mechanism to deal with noises effectively, it is necessary to recognize their nature and to classify them as regular and random, frequent and rare, etc. Generally, regular noise is the result of technical problems with measuring equipment, or interference from other devices operating in the vicinity of a magnetometer. An indicative example of regular noise is interference in the magnetic data that arises when ionosondes are operating. This is quite a common problem at integrated remote observatories, where it is physically and organizationally impossible to distance magnetic and ionospheric measurements far enough.

Figure 2 (left panel) shows hourly record from Magadan observatory obtained by fluxgate magnetometer FGE with Magdalog datalogger. It contains noises caused by ionosonde operation with regular sessions every 5 minutes. These pulses have quite large amplitude and duration and produce significant noise in the minute data obtained by averaging using the Gaussian filter (INTERMAGNET standard). A similar picture of interference caused by ionosonde during vertical sounding every 15 minutes is also visible on the daily record obtained by the similar magnetometer at Paratunka observatory, see Fig. 2 (right panel). Since the noise amplitude does not exceed 2 nT, in order to detect it, natural



variations of the field were eliminated according to the data from other magnetometers installed in the same pavilion: Z component variation was removed using the data of the fluxgate device FRG-601 (Fig. 2b), F(scal) variation –was removed using the data from scalar magnetometer GSM-90, F(var) was calculated from the FGE variations and the corresponding baseline values (Fig. 2a). The authors have also observed similar interference in the raw data from Novosibirsk (NVS, fluxgate magnetometer LEMI-008) and Yakutsk (YAK, FGE magnetometer and Magdalog recorder) observatories. The distance to ionosondes did not exceed 200-300 m.

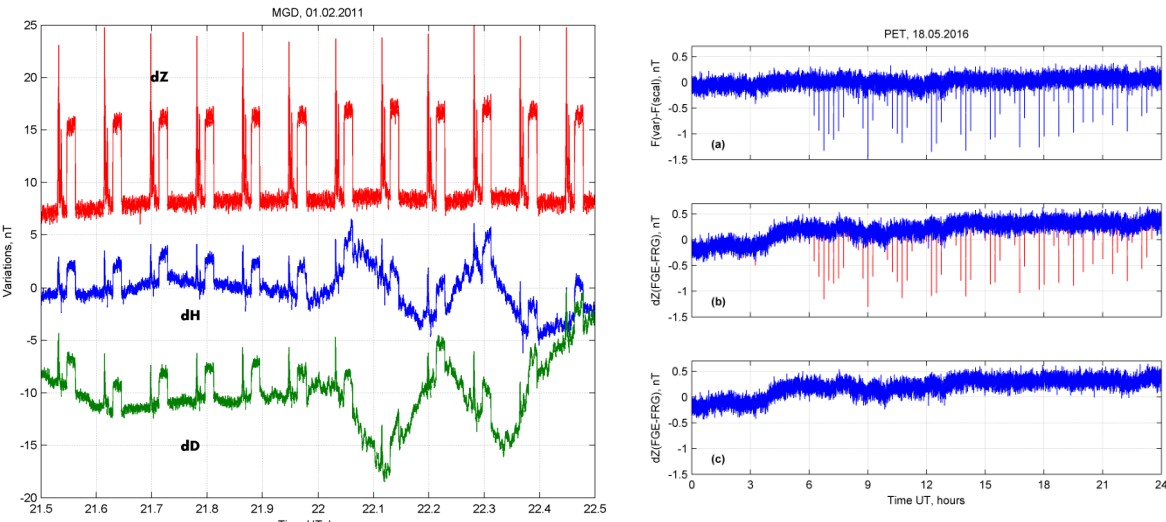

**Figure 2. Example of regular noise in magnetic data arising during ionosonde operation. - Left panel: at Magadan observatory the ionosonde operates with five-minute periodicity and makes noise signals in all components, recorded by FGE fluxgate magnetometer. Right panel: at Paratunka observatory the ionosonde operates with 15-minute periodicity. The effects appear in all records of fluxgate variometer FGE, mainly in the vertical component Z. These spikes are visible in the differences between total intensity F(var), calculated from FGE data, and F(scal) calculated from scalar magnetometer GSM-90 (a). Subplot (b) shows the differences of Z records of FGE (with noise from ionosonde) and Japanese fluxgate magnetometer FRG-601, which is not affected by ionosonde activity. Subplot (c) shows the result of clearing of the ionosonde effects from FGE records.**

In all cases, the interference was recorded by variation fluxgate magnetometers. At the same time, in the records of the other fluxgate devices located in immediate proximity, such interference from the ionosonde was not observed, for example, in the results of the fluxgate magnetometer FRG-601, the data of which were used to remove natural geomagnetic variations (it is illustrated by the difference between the data obtained by FGE and FRG magnetometers shown in Fig. 2b and c). It should be noted that the interference is regular, because it is determinated by sounding sessions, the beginning of which is



synchronized with UTC. However, small shifts are observed, which are associated with a particular mode of sounding and the range of operating frequencies of the ionosonde.

Regularity and accurate synchronization of such interference with UTC makes it easier to identify them and to make a
decision about the removal or correction. Certainly, this applies to relatively short noise signals, usually to pulses. A special software module was developed and integrated into the software package for raw magnetic data processing at MGD and PET observatories. The module generates a temporary mask, daily array of 0 and 1, where 0 corresponds to the measurements that include noises. Parameters of the "0-1" sequence are set in a special text file that contains the periodicity of sounding and the shifts of the beginning and the end of a removed interval with respect to the beginning of a sounding session for each day.
During the processing of the raw data, the mask is applied to the original daily time series and the data marked by "0" are filled with special values NaN (Not-a-Number). An example of such clearing is shown in Fig. 2c.

Interference between closely spaced magnetometers may be a source of regular noise. It is a well-known that proton magnetometers can cause interference either by the generation of additional external magnetic fields during polarization of
the proton rich liquid, or via direct influence of the DC current powering the proton (or Overhauser effect) magnetometers, which is modulated by the periodicity of measurements. For example, the effect up to 0.2 nT is expected at distance of about 5 m from proton magnetometer (Auster et al., 2008). Vector magnetometers using proton sensors in the coil systems (dIdD GSM-19FD, GEM System; POS-4, QMLab), besides the effects during polarization, also produce significant additional magnetic fields affecting the measurements of devices located in vicinity. Figure 3a and b shows manifestation of the dIdD
magnetometer operation at Cape Schmidt observatory in the records of dH and dD variations from fluxgate magnetometer MAGDAS. Due to the polar specifics (necessity of heating and the absence of additional pavilions) the both devices are installed on one pillar at a distance of about 2 meters, almost on the same meridian. MAGDAS measurement frequency is 1 Hz. Oscillations in a range up to 2 nT with a period that is  multiple of dIdD measurement periodicity (2.5 s) are observed. There is also noise in the record of vertical component. The mechanism of influence of dIdD on MAGDAS measuring
process is complex, it is associated with timer stabilites (noise amplitude "floats" over time) and practically it cannot be reliably corrected by a software during the postprocessing. Therefore, the only effective way to avoid this interference is to distance the magnetometers from each other.

Figure 3c shows the second example of the proton magnetometer influence on another device at Paratunka observatory. The
fluxgate variation magnetometer FGE signal (measuring frequency is 2 Hz) is modulated by the operation of GSM-90 Overhauser magnetometer with a measuring rate of 5 s, the range of noise in the vertical component Z is up to 1-2 nT (in other components it is less than 0.5 nT). Since the sensor GSM-90 is located at a distance of about 4 meters from the FGE sensor, the impact through additional magnetic fields during polarization is hardly probable. Presumably, the interaction takes place at the hardware or communication level, because the devices are connected by a single datalogger Magdalog. The


same noises are also observed on a similar set of magnetometers at Magadan observatory. We cannot solve this problem technically. However, the synchronicity of measurements by two instruments (on one datalogger with a single timer) has allowed us to implement a software clearing. From the fluxgate magnetometer data we select only those, which occurred during the frequency measurement of proton sensor precession, and fragments during polarization are removed.

Unfortunately, due to a little synchronicity instability of measurements by two devices, only a few samples from the 5-second cycle of FGE can be reliably distinguished (for reliability actually only one sample is chosen, which is shown in Fig. 3c by "o"), i.e. forced tenfold data loss takes place.

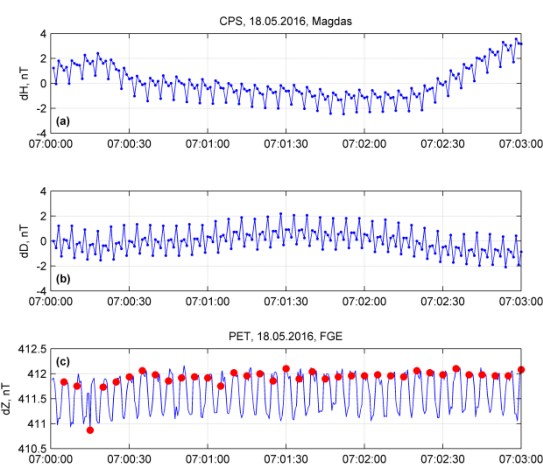

**Figure 3. An example of influence of the magnetometers with Overhauser scalar sensors: (a, b) effect of vector dIdD GSM-19FD in the records of H and D components of MAGDAS fluxgate variometer at Cape Schmidt observatory; (c) influence of the scalar magnetometer GSM-90 on the vertical component Z of the FGE fluxgate variometer at Paratunka observatory; oscillations of dZ have the period of GSM-90 measurements, undisturbed dZ are found at upper part of oscillations, red dots show undisturbed values, which are used for next processing.**

In minute values, calculated in accordance with INTERMAGNET standard, normally distributed noise can (in special cases) be removed by filtering. However, the fact that noise is often asymmetrical creates at least two problems on minute intervals:

(a) when averaging the noise with asymmetric signals, the obtained mean values are also biased. For the noise shown in Fig.
3c with magnitude up to -1 nT, the average estimate will be systematically biased downward by about 0.5 nT, this value is quite significant in relation to current requirements for long-term magnetic measurements;



(b) if during the absolute observations a reading at the zero position of DIflux magnetometer fluxgate sensor coincides with noise in the variometer, than a difference between absolute and variation measurement arises, i.e. accuracy of baseline values is decreased, and in the worst cases a systematic errors in final values of the total field vector arise.

## 3.2 Noises of different shapes

### 3.2.1 Pulse noise

A generalized view of possible shapes of structured noise is presented, for example, in Lopez-de-Lacalle (2016, Figure 1). Pulses (spikes) are perhaps one of the most common types of signals in magnetic records, which with a sufficient probability are not related with natural processes. Spike is interpreted as relatively short signal with a significant amplitude (duration is less than a few seconds or several measurements, if magnetometer has a measurement frequency from ones to tenths of Hz), with well-defined sharp leading and back edges similar in amplitude. If one of the edges is weak or is absent, then we can talk about the jump. All these properties of pulses can be used for their detection and removal. We can also note that pulses with duration of one measurement cycle are the most likely to be associated with hardware problems or interferences from nearby sources. The amplitude is also an important characteristic. Pulses with amplitude in units, tens and more of nanoteslas also have a low probability of being caused by natural sources.

Figure 4a shows an example of a daily record of the field total intensity F recorded at Cape Schmidt observatory by Overhauser magnetometer POS-1, the measurement periodicity is 3 s. For CPS observatory, problems with the stability of the power supply and the quality of grounding are known, to which POS-1 is sensitive enough. On the record, outliers with the amplitude up to 1500 nT can be seen (over 90 events). The duration of these pulses does not exceed one measurement, i.e. they have sharp edges with almost equal magnitude of leading and back fronts. Therefore, it is not difficult to identify and to locate them. The simplest algorithm of allocation of such signals by amplitude, based on the proximity of the pulse edges and its isolation, works quite effectively. The result of the program implementation of the algorithm is shown in Fig. 4b. It should be noted that in the Overhauser magnetometers POS-1, which are quite widely distributed at magnetic observatories, each record is accompanied by the estimation of measurement quality using a special parameter QMC (Quality Measurement Criterion), which value is related to the quality of the precession signal by which frequency the F is calculated, as well as qualitative estimates of measurement conditions such as signal-to-noise ratio, the duration of the precession signal, power supply voltage, etc. (POS-1 User manual, 2004; Denisov et al., 2006). Similar, although a little less informative estimations of signal quality are also performed for scalar magnetometers GSM (GSM-19 Instruction Manual, 2008, p.54). For POS-1 these qualification parameters are used in the standard software at IKIR FEB RAS observatories to estimate the quality of measurements within the processing, that increases the efficiency of simple mathematical algorithms.



It is clear, that noise in the record shown in Fig. 4a does not represent a problem for program processing, and in case of smaller quantity they can be processed manually. However, difficulties arise in the case of more irregular shape of noise-outliers, when they can not be considered as narrow isolated pulses. The examples are the noises described above, arising during ionosonde operation  (Fig. 2a and b), which are often extended in time and can have multimode structure. Efficiency

of their processing is provided by the strict repeatability.

Figure 5 shows the daily record of dH, dD, dZ variations, obtained at Hyderabad observatory using the fluxgate magnetometer FGE, measurement frequency is 2 Hz. As it can be clearly seen, there is irregular and frequent noise in the vertical component in the form of outliers (spikes) with amplitude of more than 5 nT. The observatory is located on the

territory of CSIR-NGRI Institute, within the city. The most probable reason of this noise is the metro line (above-ground) passing at a distance of 200-300 m to the south from the observatory. Figure 5c shows a 1.5-minute fragment for dZ, which shows that outliers have sufficiently definite and stable structure, a sharp leading edge and falling exponentially back edge, the total pulse duration is up to 3-6 samples, i.e. about 1-3 s.

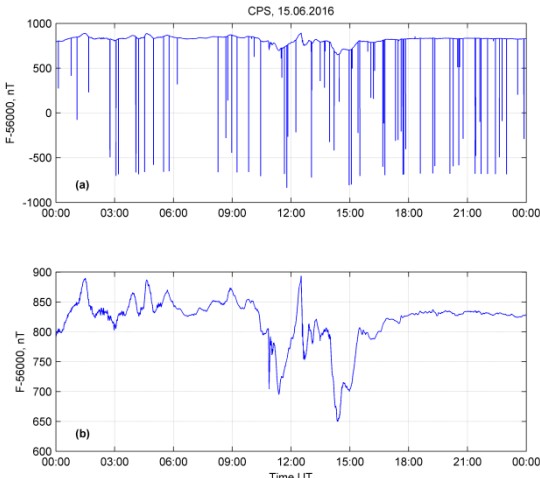

**Figure 4. An example of pulse noise in the results of measurements by Overhauser magnetometer POS-1 at Cape Schmidt observatory. (a) the total intensity F raw record, (b) signal after noise removal.**





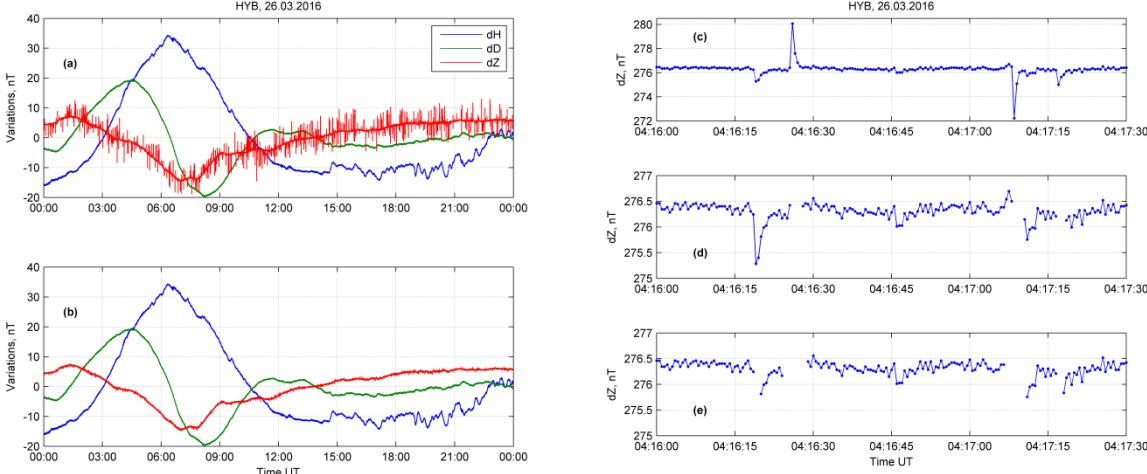

**Figure 5. Example of frequent pulse noise at Hyderabad observatory: (a) original daily records of variations dH, dD, dZ; (b) the same of (a), but after noise identification and removal in vertical component; (c) 1.5-minute fragment of daily dZ record, showing the detailed structure of the pulses; (d) results of noise clearing using a simple algorithm; (e) result of clearing using an algorithm based on wavelet transform.**

The algorithm applied at HYB observatory to detect and remove the noise caused by metro line is based on its structural stability and works as follows: when a change of dZ between neighboring measurements (in absolute value) exceeding a given threshold is found, three subsequent measurements are discarded. It is clear that this algorithm creates risks: (a) natural signals with sharp edges, for example, during magnetic disturbances, may be discarded; (b) noise may be removed not completely, if its duration is longer than given one; (c) noise can be missed, if due to a small shift its leading edge takes 2 samples. However, to calculate the minute values of the magnetic field variations, that may be enough. Figure 5d shows the results of the algorithm application during real processing of raw HYB data (threshold value dZ/dt = 1 nT/0.5 s was used).

An algorithm based on continuous wavelet transform showed higher efficiency in detection of such spikes. For the first time it was proposed in the paper (Zhizhikina et al., 2016). This algorithm includes the following main operations: (1) wavelet decomposition of data on informative scale levels (determined during algorithm construction) is performed; (2) spikes are detected on the basis of threshold functions (different thresholds for each scale level and for positive and negative values of the wavelet coefficients are used).

The algorithm efficiency is determined by the wavelet transform sensitivity to sharp changes of function values, amplitudes of wavelet coefficients significantly increase in the areas containing local features in the form of sharp peaks (Daubechies, 1992). Figure 5b and e shows the results of the algorithm.


The geomagnetic variations and noise are dependent from location of the observatory. Therefore, preliminary tuning of the algorithm parameters is required for the selected observatory. Currently, the algorithm is adapted for middle latitude Paratunka observatory and for equatorial Hyderabad observatory. The values of the parameters were defined for criterion of

the absence of false detections, using selected data. The effectiveness of the algorithm was estimated for quiet and disturbed magnetic field, the spikes detected by the experienced magnetologist were considered as reference. Table 2 shows the results of the estimation of the algorithm effectiveness for Hyderabad observatory.

Table 2. The results of the estimation of the algorithm effectiveness for Hyderabad observatory

| Magnetic field conditions | Number of spikes, detected by magnetologist | Detected spikes, % | |
|---|---|---|---|
| | | Wavelet-based algorithm | Simple algorithm |
| Quiet (local K < 3) | 1591 | 83.91 | 73.35 |
| Disturbed (local K ≥ 3 | 1495 | 85.35 | 73.85 |

The results show high reliability of the wavelet-based algorithm to detect the main part of spikes: about 99% pulses with high amplitude (> 3 nT) are isolated. But pulses with small amplitude (<0.5 nT), which are 25% from all pulses, can not be reliably detected by magnetologists. This restricts the optimization of the algorithm.

### 3.2.2 Jumps

In a certain sense, the jumps in the results of measurements can be considered as pulse noise described above, but with a continuous interval of record between the leading and the back edges or in the absence of back edge. In case of sharp edges and sufficient amplitude of a jump, it is not difficult to identify it. However, unlike spikes, in practice such jumps are quite rare in raw magnetic data (for example, Fig. 2). Changes of magnetic record level with gentle edges or with edges that contain other noises with considerable amplitude are predominantly observed. The reasons of such jumps are technical

operations with equipment, change of the magnetic state in the pavilion with the magnetometer or near it, change of instrument parameters, etc.

An example of noise in the form of jumps caused by the magnetic environment changes near the pavilions at Paratunka observatory is shown in Fig. 6. The effect is visible in the field total intensity F, which occurred during removal and

reinstallation of casing pipes in a well of 80 m in depth, located approximately at a distance of 100 m to the south from the magnetic pavilions. These operations were carried out within six hours, heavy machinery was used (truck crane, bulldozer). In general the effect does not exceed 1 nT, but it stands out well in the difference between the records of the two scalar magnetometers, located at different distances from the well (dIdD is the closest, POS-1 is the most remote, the distance between them is 30 m). Each removal of a pipe section from the well causes a jump of the magnetic field gradient between




POS-1 and dIdD by about 0.1 nT. Lowering back into the well looks like a recovery process in dF. Much smaller effect is observed in the difference of records of POS-1 and GSM-90 magnetometers, located at about the same distances from the well. This example represents the situation which is quite widespread at magnetic observatories and shows the following important points:

(a) identification of signals of small amplitude by mathematical methods of pattern recognition, even if they are rather different from the field natural variations, can be practically implemented in very rare cases (only for typical noises and in case of large samples of a priori data);

10    (b) such signals can be identified reliably only in difference data obtained by spaced magnetometers;

(c) practically, the only way to detect such signals is by an experienced, trained magnetologist, largely based on the additional information about measurement conditions.

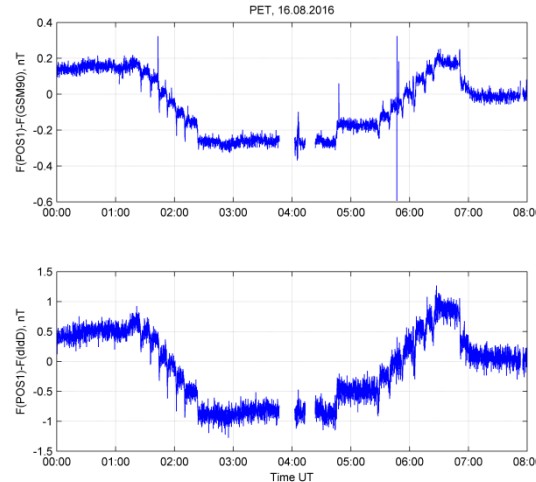

**Figure 6. Example of jump noise at Paratunka observatory during removal and lowering of casing steel pipe into the well of 80 m depth. Curves on the plots are differences of field total intensity F measured by scalar magnetometers POS-1, GSM-90 and dIdD at various distances from the well. Each step corresponds to the operation with one pipe section.**

In relation to the example given in Fig. 6, we can assert that only a magnetologist (expert) could recognize field variations as a noise, analyzing the differences of measurement results obtained by the spatially separated devices, noting artificiality of


jumps in the variations and knowing that works on the well were carried out at that time. Similar anthropogenic disturbances are practically not corrected and in most cases the record including noises is just removed.

Figure 7 shows an example of jumps in the daily record of the field total intensity F, obtained by dIdD GSM-19FD and POS-
1 magnetometers at Cape Schmidt observatory. It can be seen in Fig. 7c, that the measurement difference of these two Overhauser magnetometers contains jumps with an amplitude of several nT. However, noises arise only in the dIdD record (Fig. 7b) and in most cases have sharp edges. The magnetometers are installed in different pavilions (dIdD is in a variational, POS-1 is in absolute), at a distance of about 30 m. The situation shown in Fig. 7c is rather characteristic for the observatory and occasionally it reoccurs. The source of the noise is not defined, but, perhaps, it is associated with interference of supply
lines or communication cables in the variation pavilion or with the currents in moisture saturated soil near the dIdD.

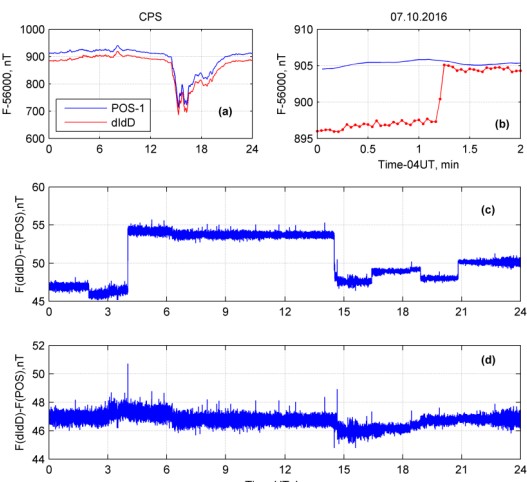

**Figure 7. Example of noise in the form of jumps in the data of dIdD GSM-19FD magnetometer at Cape Schmidt observatory. (a) daily records of F of dIdD and scalar Overhauser magnetometer POS-1; (b) detailed fragment from**
**(a) with a jump; (c) differences dF=F (dIdD)-F(POS) in which natural geomagnetic variations are removed; (d) the same as (c), but jumps in F(dIdD) are removed.**

In this example, our interest is not in the cause or mechanisms of noise, but in the possibility of its identification and correction. Since the amplitude is fairly significant, edges are sharp, and measurements are not burdened by this noise, the
posed problem can probably be effectively solved by software tools. However, in this case, processing is performed by a magnetologist, who estimates the size and the location of the jump using F(dIdI)-F(POS) plot, followed by program correction of F(dIdD) and, if necessary, removal of unreliable data at the time of the jump, if it had a significant duration. Fig. 7d shows the results of the procedure described above.




It should be noted that the jumps, after which the record level is changed and retained for a long time (several days or longer, for example, after magnetometer reinstallation), are recorded in the baseline values of variational instruments and they are eliminated during calculation of the total field vector using the standard measurement technology at magnetic observatories (see also some remarks in Sect. 4). Of course, such a method works with rather frequent absolute observations and usually

additionally requires compensation of the jump in the part of the daily data due to the fact that according to the standards, the baselines are taken as constants within a day. In any case, the undisturbed magnetic records are needed as references.

### 3.2.3 Bay-like noise

Bays are a common type of noise at magnetic observatories.  They are often the result of changes in the magnetic field near the magnetometer due to moving objects with magnetic effect, for example, a car, a person with instruments, etc. In case of

such noise, shapes of signals in the field components are specific and connected among themselves. Nevertheless, this noise, if its amplitude does not reach extreme values, is hardly distinguishable from natural variations. The possible ways of identification are comparison with the data obtained by other magnetometers (gradiometer principle) and analysis of the information about events at an observatory (logging of such events is the direct responsibility of the observatory and its staff).

Figure 8 shows bay-like noise occurring during off-roader driving near the magnetic pavilions of Cape Schmidt observatory and during its return. Variations of dH, dD and dZ components are obtained by MAGDAS magnetometer, variations of field total intensity F are obtained using dIdD and POS-1 magnetometers (Overhauser sensors). MAGDAS and dIdD are located in a variation pavilion, POS-1 is installed in an absolute pavilion. The distance between the pavilions is about 30 m. Noise

duration is about 30-40 s, the amplitude is up to 20-30 nT, their time shift is clearly defined. We may also note the dependence mentioned above between the form of a signal in different field components. The identification of a signal as the noise was made by a magnetologist by the shape of a signal, time shift and by comparing it with similar signals that have been earlier observed (in this case there was no information about the source of the noise at the observatory). After localization, the noise was removed by a magnetologist during raw data processing.

One more example is shown in Fig. 9. A bulldozer cleared the way to magnetic pavilions from snow at Khabarovsk observatory. Field variations were recored by the quartz magnetometer CAIS. Noise duration is about one minute, amplitude is maximum in the vertical component (up to 10 nT). It would be difficult for the staff of the observatory to identify  the signal in Fig. 9 as noise without the information about the works conducted near the pavilions, because according the signal

parameters, it is rather close to natural geomagnetic variations (an exception is possible only for Z-component, since its natural variations on this day did not exceed several of nT).  Just like in the previous case, the noise was removed by a magnetologist during raw data processing.





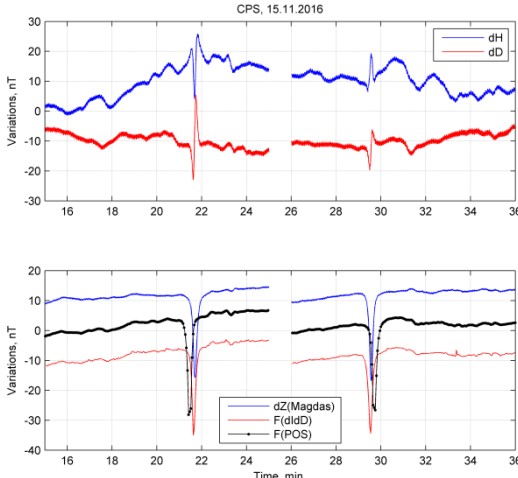

**Figure 8. Noise caused by a vehicle driving near the magnetic pavilions at Cape Schmidt observatory (in one direction and back). Upper panel: dH (upper curve) and dD (lower curve) variations, recorded by MAGDAS magnetometer.**
5 **Lower panel: dZ (MAGDAS, upper curve), dF (POS-1, middle curve with dots) and dF (dIdD, lower curve). MAGDAS and dIdD magnetometers are located at a distance of about 30 m from the POS-1.**

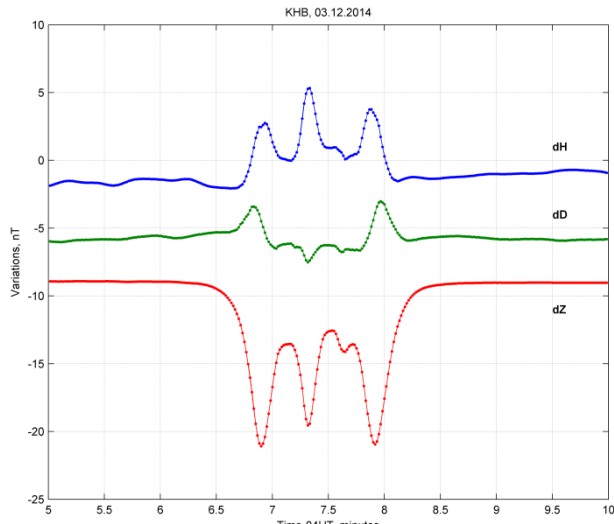

**Figure 9. Noise caused by a bulldozer clearing the way to the pavilions from snow at Khabarovsk observatory.**
10 **Variations dH, dD and dZ were recorded by quartz magnetometer CAIS. Oscillations on the records are the result of the tractor work at a distance of about 30 meters from the magnetometer.**


### 3.2.4 Random-like noise

Man-made disturbances, which are expressed as additional random noise at the background of original useful signal will be understood as random noise. This notion generalizes a very big class of noises which are often difficult to classify. In many cases these noise is not localized in frequency and/or time domain. They may have a hardware origin or may be connected

with real field noisiness from external sources at the observatory. Almost always the problems of these noises are solved either technically (fine adjustement of magnetometers, improvement of grounding, power supply, etc.), or organizationally (moving of a measurement point, replacement of a magnetometer by another one which is less sensitive to noise, etc.). It is hardly probable to find effective methodological and program approaches.

Identification of hardware noise in data is possible if we compare the results of measurements by different magnetometers or if we change the operating modes of a device if it is the only one at the observatory and if it is possible by the specification of a devices. The increased noise of the magnetic field at the observatory (as a result of total impact of  many factors) can be identified, for example, if we make measurements by the same magnetometer at the observatory and in the place with obviously low noise. In general, assessment of the background noise is a labor-consuming task including a research aspect

and very often it has not effective practical results. The problem becomes more complicated due to the fact that in many cases it is almost impossible to distinguish the noise background from a natural signal which is a subject of scientific research, for example, seismomagnetic effects. Fig. 10 and 11, as an example, show the random noise which arises in the results of magnetic measurements in case of problems with power supply.

Figure 10 shows the results of daily measurements of the field total intensity F, made at Khabarovsk observatory using scalar Overhauser magnetometer POS-1. There was a general power outage at the observatory and in the nearby settlements at about 08:50 UT. The measurements were continued using an autonomous power supply system at the observatory. At about 12 UT the external power supply was restored. On the record obtained by POS-1 noise with the amplitude up to 1 nT was recorded. It lasted for more than a day. For a descriptive graphical representation, natural geomagnetic variations were

excluded using the data from another scalar magnetometer (GSM-19W), in the measurements of which the noise caused by power failure did not manifest  (Fig. 10a).  Fig. 10b shows the behavior of QMC parameter, which estimates the quality of the precession signal of POS-1 Overhauser sensor in nT units and was  described in Sect. 3.2.1. In this example, the fact that the noise remained in the measurements of POS-1 after the restoration of external power is of interest, i.e. the fact of presence of the noise could not be ascertained according to the staff information about the situation at the observatory, it

could be determined only by the direct visual analysis of these measurements. And the second fact is that in this case an effective way to recognize the noise is to estimate the behavior of QMC parameter.



The second example (Fig. 11) is appearance of noise in the data of dIdD GSM-19FD magnetometer, registered at Karymshina station of IKIR FEB RAS. Karymshina station is located approximately 15 km from Paratunka observatory, a place with a minimum of possible industrial sources of noise, including the absence of external power supply by power lines. At about 02:45UT a failure of a diesel generator occurred and emergency scheme of power supply with external battery

package as a source of voltage and disconnection of all powerful devices has been activated, the standard power supply was restored at about 11:55UT. It is shown in Fig. 11a that pulses with amplitude up to 10-20 nT and random noise up to 5 nT appeared during the operation of the emergency power system in two measurement channels of dIdD (with additional fields in coil systems D and I, see, for example dIdD Instruction Manual (2010)). For illustration purposes, Fig. 11b shows the record after eliminating the low-frequency variations. Since the amplitude of the noise is significant, it can be easily

identified during the analysis.

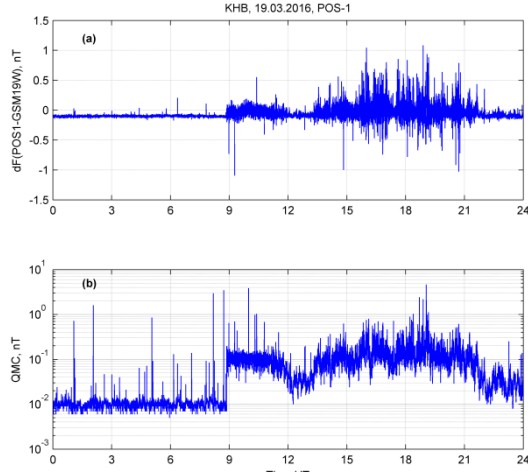

**Figure 10. Noise-like interference caused by the problems with power supply at Khabarovsk observatory, manifested in the records of Overhauser magnetometer POS-1. (a) daily record of F(POS-1) after elimination of natural geomagnetic variations by GSM-19W data; (b) behavior of QMC (Quality Measurement Criterion) instrumental**

**parameter of POS-1.**





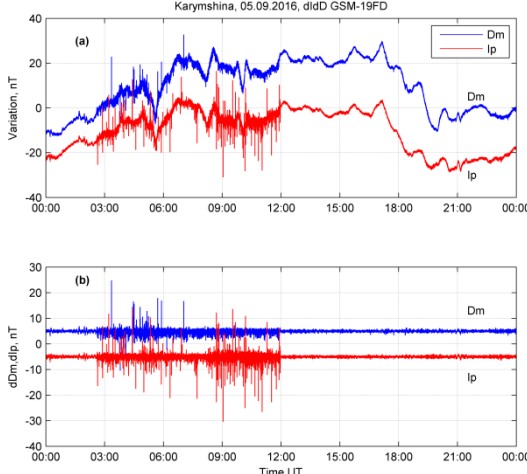

**Figure 11. The noises in dIdD data which arose due to the failure of a power supply system at Karymshina station (15 km from Paratunka observatory). (a) the field total intensity F (Dm, Ip) obtained in case of additional fields in the coil system; (b) the same as in (a), but slow variations are excluded (for clarity).**

In both examples given above, the only way to exclude noise is to remove all data fragments which contain noise. It is impossible to restore the original useful signal with the acceptable quality.

### 3.3 Critical and weak noises

This criterion is important due to the fact that it determines the extent of noise influence in magnetic data on the results of researches which are carried out using these data. When a researcher uses the final data from observatories published in some way, in most cases he has no information in what conditions the measurements were performed, what procedures were applied during preliminary processing (except for the cases when these procedures are prescribed by standards), etc. Thus, the responsibility for the quality of the data provided to scientific community is very high and it completely lies on an observatory.

Let us consider a "classic" case. An observatory of INTERMAGNET network obtains primary 1-second data of dHs, dDs, dZs, Fs variations by direct measurements and then, using baseline values, calculates total Hs, Ds, Zs, Fs, reduced to main pillar, and minute values Hm, Dm, Zm, Fm using the procedure, defined by INTERMAGNET standards (INTERMAGNET Tech. Ref. Manual, 2012). Filtering by Gaussian filter is quite an effective method to suppress random noise. Also Gaussian filtering works acceptably with pulses of small amplitude, but it is not effective for jumps and bays. Thus, the residual effects from strong noise are included into the final minute data, at the same time they are very smoothed, i.e. they are almost indistinguishable against the background of natural variations, but they influence the results of further calculations



performed with these data. That makes it necessary to identify these critical noises and to remove them during the primary data processing. It entails secondary problems, such as gaps in original data, which are used to calculate published minute values. In spite of INTERMAGNET recommendation, that mean values should be calculated in accordance with the 90% availability rule, such criteria are difficult to define and sometimes they are not applicable at all.

**3.4 Noise with known and unknown sources**

If it is known that a signal, which is suspected as a noise, is the result of some reasons which are not associated with natural variations of magnetic field, then it is rather a powerful argument to remove the signal. At the same time, however, there are some fine points:

(a) noise removal leads to the gaps in published data, therefore scientist in many cases before application of methods of analysis is forced to fill these gaps with dummy data, i.e. the data, calculated from available dataset using some interpolation method. Errors of calculation results arising due to the filling may be comparable or even greater than the errors that occur due to the noise which was not deleted;

(b) problems which are solved using the final data should be defined. In some cases, signals identified by the criterion of origin as noise can be of independent scientific interest. For example, in practice of observatory measurements, a fictitious "seismomagnetic effect" is well-known, when we observe oscillations in the data of magnetometers with a suspended system for compensation of sensor inclinations or in the records of induction magnetometers. These oscillations arise when a seismic wave from a near or strong earthquake is passing the place where the magnetometer is installed. As an example, figure 12 shows the records of magnetic field variations at IKIR FEB RAS observatories . On these fragments the effect of a strong earthquake with the magnitude of 8.3 which occurred on 24 May 2013 in the Sea of Okhotsk at the depth of about 600 km is illustrated. The following magnetometers were applied: dIdD GSM-19FD (Dm and Im modules with additional fields of coil system) at the CPS and PET, fluxgate FGE (H channel) at MGD, digital magnetometer with Bobrov's quartz sensors (H channel) at KHB. It is clear that the earthquake is manifested well at long distances,  the magnetometer data based on different measurement principles. For the tasks of studying the variability of the magnetic field, the signals shown in Fig. 12 are noise and they should be removed. However, if, for example, seismomagnetic effects are investigated, then the recorded "fictitious" signals in Fig. 12 would be a good bench mark to estimate the passage of a seismic wave in the area where magnetometers are installed. Unfortunately, it is necessary to note that in some cases, the researchers of seismic effects in the magnetic field even do not ask what type of a magnetometer is, the data from which they use.



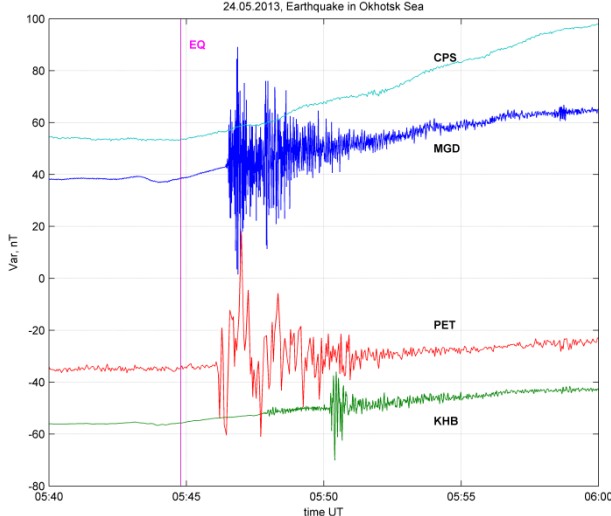

**Figure 12. "Fictitious" variations of the magnetic field recorded by magnetometers with suspension system of measuring sensors: dIdD with scalar sensor in the coil system, FGE with fluxgate sensors and magnetometer "Quartz-6" with Bobrov's quartz sensors. Records obtained at CPS, MGS, PET and KHB observatories during a strong earthquake in the Okhotsk Sea on 24 May 2013 are shown. Vertical line "EQ" shows the earthquake time.**

If a signal is suspected as a noise, but its structure does not allow us to recognize it as a noise, and there is no additional information about a possible source, then it is very complicated for a magnetologist to make a decision. In most cases, these signals are not removed, that creates risks of reducing the quality of observatory data.

**4 Possible methods for noise removal from raw magnetic data**

In the previous section we classified noises, which are encountered the most frequently in actual magnetic measurements at the observatories of IKIR FEB RAS and CSIR-NGR, and illustrated them by some characteristic examples. Naturally, the description and the samples are quite limited, since the variety of noises is extremely large.

It is clear that identification of noise is the solution only of a part of the problem. The second part is to choose an effective method of further work with this noise. Unfortunately, the choice of possibilities is very small. Mainly a fragment is simply removed from a record that results in a gap, which can be filled by the most suitable "dummy" data, or if the structure of the noise is recognized, it is removed and the original useful signal remains.

Earlier, in the description of pulses (see Fig. 2, 4 and 5), possible techniques of dealing with these noises, using automatic removal of a record fragment after identification of noise, have already been shown. In other cases, noise can be removed





manually by a magnetologist. There are different approaches to implement these procedures. If a file with raw data has text format (for example, POS-1 User manual (2004) and dIdD Instruction Manual (2010), it is possible to remove the unreliable records manually. In this case, during the further work with the corrected file, there will be gaps not only in the measured magnetic values, but also in a timestamp, that is not always convenient if we need a uniform time grid. Another possible

approach is to create an intermediate file, usually in text format, in which noisy data are manually or semi-automatically replaced by the values that indicate unreliable measurements, for example, "99999". This option is particularly useful if the initial measurements are recorded in files in binary format, e.g., MAGDAS-A Installation Manual (2005). In this case special converter software is usually used, including those which allow an operator to encode the required data as unreliable.

In the software for magnetologists developed in IKIR FEB RAS, which is based on MATLAB and Octave mathematical packages (Khomutov, 2016), the third approach is used. It is based on the following methodical principles: (a) original files are always used in processing of raw (original) magnetic data; (b) the intermediate files are not formed, i.e. through stream processing is performed from the raw data to the required final result, all the results of intermediate computations remain only in the computer's random access memory. Information about unreliable data is stored in special text files, as a date, start

time and end time of the interval, which is required to be removed and special features, for example, for the choice of the magnetic components and for comments. An example of such a record for the FGE magnetometer at Paratunka observatory is shown below.

```
%  Date       UT1     UT2    HDZ
2016 01 30 03.4272 03.5211 100 % earthquake
2016 01 30 03.4269 03.5795 010 % - " -
2016 01 30 03.4274 03.4973 001 % - " -
2016 01 30 03.7199 03.7283 010 % aftershock
```

25 Maximally simplified text format is used, that reduces the probability of errors during manual typing and speeds up the file reading. The example shows the information, used to remove the noise caused by a nearby earthquake on 30 January 2016 in the data from FGE magnetometer. In the processing of measurement results of 30 January 2016, D values from 03:25:37(=03.4269) to 03:34:46(=03.5795)UT, which are noisy due to the mechanical influence of a seismic wave on the suspended sensor, will be filled with a special symbol NaN and will be excluded from the further processing. The boundaries

30 of intervals with noise are defined by a magnetologist in interactive mode by record plots for the corresponding components and with required scaling that is provided by convenient interface with graphics in MATLAB and Octave. Noise in other components is removed similarly. The effect of the described earthquake and the results of applying the clearing procedure are shown in Fig. 13.




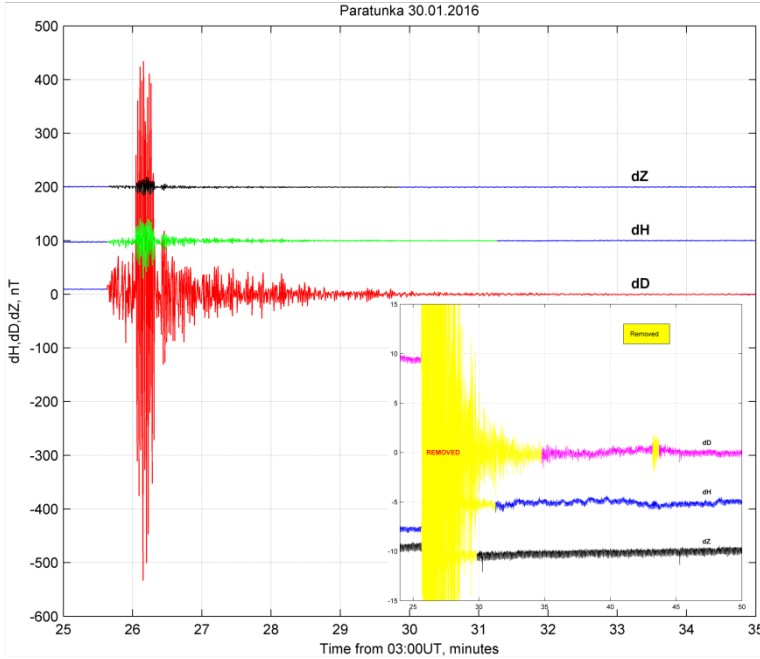

**Figure 13. Example of manifestation of a nearby earthquake on 30 January 2016 (magnitude 5.7, distance is about 100 km) in the record of FGE magnetometer with a suspended system at Paratunka observatory. The inset shows the results of data clearing performed by a magnetologist using a semi-automatic procedure (it is shown by yellow). In component D the effect of an aftershock, which occurred in 17 minutes, is also removed. Jumps of average level of the records, reaching 10 nT in D (about 1.5') are clearly visible in the inset.**

Considerably a more difficult situation arises in the attempt to restore an original signal, i.e. removal only of noise from the measurement results. It is impossible in the most cases in actual working practice of the observatory due to the unknown structure of a useful signal and the unknown structure of noise. Analysis of a concrete situation is individual, and often grows almost to a scientific research. However, in some simple cases the problem can be solved. These situations include, for example, noise, leading to jumps of record level, sometimes followed by restoration also in the form of a jump. It is clear that in this case the noise is a constant addition to the useful signal, and after its subtraction, the initial undisturbed signal will be restored. The main problem is the correct estimation of the value of the noise (magnitude of a jump) and the necessity to be sure that noise has constant value in the analyzed interval of the record.

In practice, identification of a jump and estimation of its parameters (time and magnitude) is performed by a magnetologist, usually in interactive mode of work with magnetic record graphs. If the jump was very rapid and time of a transitional process is comparable with the duration of the interval between samples, and generally faster than possible natural variations in the magnetic field, then the jump parameters can be estimated visually with sufficient reliability. If transitional process is



long enough, then field natural variations can significantly affect the accuracy of the estimation. In this case, it may be useful to compare the analyzed record with measurement results obtained by another magnetometer, because natural geomagnetic variation will be excluded from the difference between two records, and the jump will be manifested as two permanent levels of the difference. However, in this case, there are also a lot of limitations, for example, the nature of the cause of the jump. If

the jump is caused by a change in the magnetic field, the source of this change is fairly close to the place of measurements, and the compared magnetometers are located closely (usually in the same or in neighboring pavilions), then noise effect will be manifested in the data of both devices and it will be difficult to make any reliable estimations. If the reason of the jump is of technical origin or its impact on the supporting magnetometer is small, it is possible to estimate the required parameters.

The situation described above is partly similar to that which occurs in case of elimination of long-term changes of the level of magnetic record from variation magnetometer, using absolute observations. Figure 14 shows an example of compensation of jumps that arose in the records of FGE magnetometer at Paratunka observatory after the earthquake on 30 January 2016 (see. Fig. 13). Regular absolute magnetic observations and dense series of obtained baseline values allow us to eliminate effectively the effect of such jumps in the series of full values of the magnetic field components, i.e., for example, the total

values of the declination $D = D0+dD$ since 31 January 2016 will be free of the jump effects in case of earthquake.

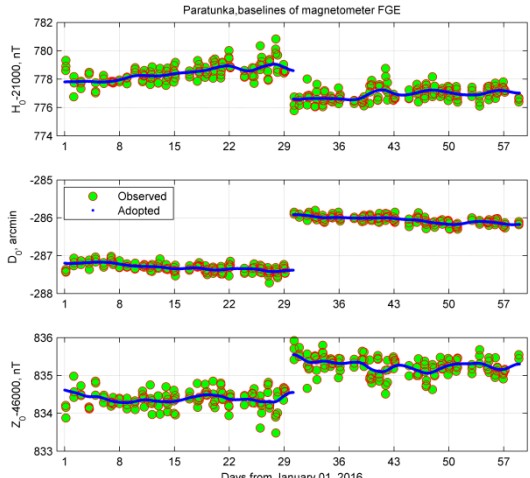

**Figure 14. Baseline values of FGE variation magnetometer, Paratunka observatory, from January to February 2016. Marker "o" shows individual baseline values (Observed), continuous curve shows the values adopted for each minute**

**(Adopted). The jumps in baselines due to the earthquake on 30 January 2016 are clearly visible.**



## 5 Conclusions

The review of noises in raw magnetic data and some methods of its identification and removal given above represent only a part of the real situation with which a magnetologist deal at an observatory when processing the measurement results. However, even with such a volume it is possible to draw the following important conclusions:

1) in most cases, the correct identification of noise can be performed only according to the estimates made by an expert, a magnetologist of the observatory, who uses raw (original) results of measurements and all available additional information about measurement conditions;

2) the most important source of information about noise is the comparison of measurement results obtained by different magnetometers, including those using different measuring principles, as well as careful monitoring of the environment at the observatory;

3) automatic program identification and correction of noise have an auxiliary nature and are principally an interactive tools
to help a magnetologist in data processing.

All this indicates the fact that data processing should be carried out by a qualified magnetologists directly at a workplace, i.e. at the observatory, and include the whole set of requirements (Jankowski and Sucksdorff, 1996; Nechaev, 2006; INTERMAGNET Tech. Ref. Manual, 2012). The similar opinion is presented by Linthe et al. (2012). Unfortunately, at the present time, many observatories, especially newly created and located in remote areas, have problems with staff and their
qualification. In these cases it seems reasonable to create centers for collecting raw magnetic data, where full cycle of data processing would be performed. Examples of such centers are BGS with a center in Edinburgh (http://www.geomag.bgs.ac.uk/data_service/space_weather/current_conditions.html) or GC RAS (http://geomag.gcras.ru/) which collects and processes the data from the Russian magnetic observatories. At the same time, a part of the problems, including those connected with incorrect noise processing can be solved ineffectively.

*Competing interests.* K. Aurora is a member of the editorial board of the Special issue of journal.

*Acknowledgements.* The authors are grateful to the staff of observatories of IKIR FEB RAS for providing qualitative
magnetic measurements. Stanislav Nechaev, main magnetologist of Patrony observatory (Irkutsk, IRT), Pavel Borodin, magnetologist of Arti observatory (Arti, Ekaterinburg, ARS), Zinaida Dumbrava, Head of Khabarovsk observatory (IKIR FEB RAS, KHB) and Vladimir Sapunov, Head of Quantum Magnetic Laboratory (Ural Federal University, Ekaterinburg) are acknowledged for helpful many-year discussions, which improved our understanding of noise in magnetic observations.





Phani Chandrasekhar and K.Chandashakhar Rao are acknowledged for their contributions for Observatories in Hyderabad and Choutuppal. Director of CSIR-NGRI is acknowledged for his permission to publish this work. The DST-RFBR collaboration is acknowledged for funding the joint studies (Grant of DST No.INT/RUS/RFBR/P-234, dated 28-9-2016 and Grant of RFBR No.16-55-45007). Magnetometers POS-1 and POS-4 for the observatories of IKIR FEB RAS were

purchased with grant funds of the Russian Science Foundation, project number 14-11-00194. The authors are very grateful to Anna Larionova for her help in English correction, to the referees and the Editorial Advisor for the comments and corrections made in the manuscript. They greatly improved the quality of presentation.

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
