# Peer review of "Noise in raw data of magnetic observatories"

_Geoscientific Instrumentation, Methods and Data Systems, 2017_

## Referee Comment (RC1) · Anonymous Referee #1 · 30 May 2017

General comments:

The content is relevant and the paper is largely organised in a logical way.

The paper needs considerable language improvement to grammar and style level as well as conciseness. I don't recommend publication until the authors have been working significantly on this. Pointing out all the necessary language corrections is beyond this review, and the list of special comments is addressing few of them. The manuscript needs careful attention, preferably from a native speaker. Some shortening should be possible in this process without loss of information.

The classification of types of noise presented in this paper includes some very special

terms, which are not very common. Examples are 'regular-random noise', pulse-noise', These are not common expressions and should be better explained in the text or replaced by more common expressions like 'spike' if possible.

Special comments:

I think the word 'magnetologist' could be more clearly defined on page 2, line 11, as it is used throughout the text.

You never used time derivatives of H, D, Z to visually identify small spikes. Why not? I think this is a very useful method to identify small artificial disturbances in the presence of larger natural field changes when you only have one recording instrument. It acts like a low pass filtering.

The term 'noise' is singular; please don't write 'noises'. Either 'noise' or, if you want to express plural, use terms like 'types of noise' or 'forms of noise'.

page 1 line 11 word centres -> World Data Centres

l. 12 primary results -> raw data?

l. 19 upper shells -> lithosphere? near Earth space?

l. 22 to 27 Not sure what is meant and I likely disagree: by giving rules for variational data, INTERMAGNET is effectively giving rules raw data. Every observatory has to submit 1 year of data (minute means from variometer) to become accepted as INTER-MAGNET observatory. In this process, the data is checked by experts. Possibly, the authors want to address mean minute means (variational data) and higher resolution spot readings (raw data)? This is now addressed in 1-second data of INTERMAGNET.

l. 27 What is primary magnetic measuremtns?

page 2 l. 5 noises -> noise registered -> recorded

l. 11 oriented to -> targeted at OR aimed at

[Figure]

page 3 l. 5 conventionallyconsidered -> conventionally considered

Table 1 Please introduce space between the column for longitude and the column for institute. Please give a short description of the instruments used, especially when such information is not available from a current vendor website or manual. I personally don't know the FGR-601, the Quartz-06 (Bobrov-type?), asn the difference between POS-1 and POS-4.

p. 9 l. 10 Does 'well-defined sharp leading and back edges' not also describe the features in Fig. 2 pretty well? But noise in Fig. 2 is supposed to be very different from noise in Fig. 4, I don't see the difference here very clearly. Please improve argumentation and possibly structure of the paper.

l. 13 amplitude in units, tens and more of nanotesla -> amplitudes of tens of nanotesla or more

l. 21-22 This sentence is very difficult to understand. Please reword. This is just one example of a sentence that has to be reworded to become understandable. Please check your manuscript and make sure it contains only sentences that can be clearly understood.

l. 25 This sentence is very difficult to understand. Please reword. This is just one example of a sentence that has to be reworded to become understandable. Please check your manuscript and make sure it contains only sentences that can be clearly understood.

p. 10 l.5 This sentence is very difficult to understand. Please reword. This is just one example of a sentence that has to be reworded to become understandable. Please check your manuscript and make sure it contains only sentences that can be clearly understood.

Figure 5: Why is only the Z-component affected. Can this be explained by the source of the disturbance? (Noise in the Z-component indicates that there is a horizontal

cable with DC somewhere, noise in the horizontal components indicates that there is a current sheet in the ground below the instrument.).

p. 12 l.1 dependent from -> dependent on the

l. 13, l. 20 This sentence is very difficult to understand. Please reword. This is just one example of a sentence that has to be reworded to become understandable. Please check your manuscript and make sure it contains only sentences that can be clearly understood. l. 23 caused by the magnetic environment changes -> caused by changes in the magnetic environment

p. 13 l. 10 spaced -> separated

l. 16 jump noise -> jumps into the well -> into a well

p. 14 l1 and 2 Similar anthropogenic disturbances are practically not corrected ...including noises is just removed -> Such anthropogenic disturbances are usually not corrected ...including the noise is removed.

p. 15 l. 5 This sentence is very difficult to understand. Please reword. This is just one example of a sentence that has to be reworded to become understandable. This is a typical example for a sentence which can be understood if you are familiar with the concept beforehand. But I think it is very difficult to understand the meaning from scratch. In this situation, the wording should be more precise.

l. 8 I don't like the expression 'bay' here at is normally used for the natural signal of the polar electrojet in auroral and subauroral stations, and not for man-made disturbances. Please use 'bay-like noise' or use a completely different term.

l. 10 What does this mean: 'are specific and connected among themselves'?

l. 20 What does this mean: 'their time shift is clearly defined'?

l. 21 and 22 Please note that in general for mid and high latitude on the northern hemisphere, magnetic field sources that move in the horizontal plane around the magnetometer are giving strong negative disturbances in Z while they give disturbances in both direction in the horizontal components. This supports your argument.

p. 17 l. 1 I am not sure why this phenomena is called 'randol-like' noise. Figure 9 I would call artificially disturbed, while Fig. 10 I would call 'noisy'.

p. 18 l. 12 problems with the -> faulty power supply

p. 19 l. 10 researches which are -> the research that is l. 10 to 11 remove 'published in some way'

l. 13 to scientific -> to the scientific lies on an -> lies on the

l. 16 of INTERMAGNET -> of the INTERMAGNET

l. 17 reduced to main -> reduced to the main

l. 18 procedure, defined -> procedure defined

l. 20 bays -> bay-like artificial disturbances

p. 20 l. 1 these critical noises and to remove them -> such critical noise and to remove it

l. 3 of INTERMAGNET recommendation, that -> of the INTERMAGNET recommendation that

l. 4 What does the term 'criteria' refer to?

l. 6 some reasons -> some sources

l. 15, 1st sentence please rewrite this sentence

l. 27 benchmark

l. 29 please rewrite this sentence

p. 21 l. 11 to 13 This is very good information. I think this information should be moved

to the front of part 3.

p. 22 l. 2 missing bracket ')'

p. 25 l. 14 Automatic program identification and correction of noise have an -> Automatic identification and correction of noise by computer programs are of

l. 19 The similar -> A similar

l. 25 can be solved ineffectively -> can become more difficult to be solved

l. 33 many-year discussion -> discussions over many years

This list of language corrections is not exhaustive, the authors will have to work through the manuscript themselves.

---

## Referee Comment (RC2) · C. Turbitt (Referee) · 17 Jul 2017

Page 7, line 17: (Auster et al., 2008) should read (Auster et al., 2007)
* * *

---

## Author Comment (AC1) · 30 Jul 2017

Dear colleague,

we thank you for attention to our work and useful comments. Please, find our answer in file gi-2017-10-RC1_Answer_20170727.pdf and text with trace of changes in file gi-2017-10_version_20170727_RC1_RC2_with_review_mode.doc.

Regards, Sergey Khomutov

Please also note the supplement to this comment:
https://www.geosci-instrum-method-data-syst-discuss.net/gi-2017-10/gi-2017-10-AC1-supplement.zip

---

## Author Comment (AC2) · 30 Jul 2017

Dear Chris,

we thank you for your comment. Please, find our answer in file gi-2017-10-RC2_Answer_20170727.pdf and text with trace of changes in file gi-2017-10_version_20170727_RC1_RC2_with_review_mode.doc.

Regards, Sergey Khomutov

Please also note the supplement to this comment:
https://www.geosci-instrum-method-data-syst-discuss.net/gi-2017-10/gi-2017-10-AC2-supplement.zip